# A Self-Narrative Study: Changes in Physical Ability and Social Communication in Children with Autism through Taekwondo Training with Elements of Music Therapy from the Parents’ Perspective

**DOI:** 10.3390/bs14070530

**Published:** 2024-06-25

**Authors:** Kam-Ming Mok, Corliss H. H. Sze, Clare C. W. Yu, Emma Mak, Dorothy F. Y. Chan, Simpson W. L. Wong

**Affiliations:** 1School of Interdisciplinary Studies, Lingnan University, Hong Kong, China; 2Department of Rehabilitation Sciences, The Hong Kong Polytechnic University, Hong Kong, China; corlisssze@link.cuhk.edu.hk (C.H.H.S.); emma.mak@iws.edu (E.M.); 3Independent Researcher, Hong Kong, China; 4Department of Paediatrics, Faculty of Medicine, The Chinese University of Hong Kong, Hong Kong, China; dorothychan@cuhk.edu.hk; 5Center for Psychiatry and Mental Health, Wolfson Institute of Population Health, Queen Mary University of London, London EC1M 6BQ, UK; simpson.wong@qmul.ac.uk

**Keywords:** cognitive ability, mental health, children, physical health, autism spectrum disorder, social skills, Taekwondo intervention

## Abstract

Autism spectrum disorder (ASD) significantly affects social and motor skills development in early, middle or even late childhood. To promote social and motor skills development among autistic children, an intervention consisting of Taekwondo and elements of music therapy was designed and implemented in Hong Kong. The objective of the current study is to investigate the effectiveness of this training, based on parents’ observations of their children who had completed the key stage of the training. Thirteen parents or caregivers (n = 13) whose children with autism participated in our Taekwondo training for two months were interviewed. Three major themes were identified by thematic analysis: (A) social skills-related change after the intervention, (B) motor skills-related change after the intervention, (C) characteristics of the intervention (i.e., session arrangement, tutor/coach attitude). Our findings show that our training was perceived by parents to have a significant impact on the promotion of physical ability and social communication in autistic children.

## 1. Introduction

Autism spectrum disorder (ASD) is a developmental disorder that hinders the development of social communication skills. More recently, research has shown that individuals with ASD experience difficulties in coordinating body movements [1]. In 2021, the Hong Kong General Household Survey found that the prevalence rate of ASD among students aged 15 or below in Hong Kong was 1.4%, with more than 11,000 children who were suffering from ASD locally [2]. Autistic children have been reported to exhibit varying degrees of social skill deficits, which can impact psychosocial development [3]. For example, several studies had shown that autistic children had underdeveloped social skills, which causes problems in a range of social behaviors, such as social reciprocity, social participation and social interaction [1,2,3,4,5,6,7]. Young ASD patients face additional challenges as they have difficulties in conversing their own emotional needs to others as well as understanding social cues elicited from other people [3]. Social competence, defined as effective interaction between oneself and others, is generally inferior in children with ASD [8]. Autistic children who have underdeveloped social skills tend to have limited social interaction with their peers [3]. Social skill deficits in early ages may have a profound long-term effect on friendship development. In one study, autistic children displayed difficulties in connecting with their peers and experienced minimal social participation and social interaction [5,6]. Apart from verbal cues, autistic children are also hyper-sensitive or hypo-sensitive to non-verbal socializing cues such as physical touching; such condition further hinders their social experiences and interaction with others [7]. As suggested in the Erikson’s psychosocial theory, social interaction is indispensable during the development of children’s self identity [4]. Without adequate social skills and the opportunity to interact with peers, autistic children have a higher chance of experiencing slower psychosocial development than their same-age counterparts [4].

Aside from social skills deficits, autistic children often face challenges with motor coordination and balancing. For example, researchers found that these children had more difficulties in maintaining balance in terms of body postural and movement coordination [9]. Limitations in gross and fine motor skills may lead to delayed motor development in autistic children [10]. Participating in physical activities is a direct way to nurture and enhance gross and fine motor skills. In a recent study, performing regular exercises in physical education classes could reduce motor abnormalities in autistic children, tapped with the Test of Gross Motor Development Second Edition (TGMD-2) [11]. Another study also highlighted the importance of regular physical activities in addressing motor abnormalities in children with autism-like characteristics. This was achieved through a school-based fundamental-motor-skill intervention that included underhand rolling, overhand throwing and other basic movement training [12].

To cater to the needs of autistic children, a sport that enhances social interaction opportunities and techniques for improving motor skills would be ideal. Taekwondo, a traditional Korean martial art, requires various movements, including strikes, kicks and blocks. These techniques, especially kicks and punches require a high level of physical coordination and control as many moves demand balancing on one side of the body; the constant practice of these moves is likely to promote daily life gross motor functioning which can help autistic children to create their body control and coordination abilities. Previous research had suggested that Taekwondo training enhances balance control in autistic children compared to those who did not receive any intervention [13]. Moreover, one of the key aspects of Taekwondo is the emphasis on following instructions, discipline, and maintaining self-control, which may enhance children’s social abilities [14]. The rules-bounded nature of Taekwondo training gives autistic children the opportunity to understand and practice social rules, which can help them make strides in their social behaviors [15]. The discipline and self-control required in Taekwondo can also help children with autism manage their behaviors more effectively, leading to advancements in their social intelligence and making connections with peers. Meanwhile, improving motor skills has provided a solid ground for autistic children to improve their socio-communication skills, promoting and generalizing the skills learned in therapeutic settings [16]. It also summarized that the effectiveness of individual sports in improving motor and social skills was demonstrated in the ASD population [16].

Engagement level is one of the concerns of our intervention design, as autistic children may lose focus and attention easily during lessons [17]. To engage autistic children during the Taekwondo lesson, an additional arrangement is needed. Music has been found to be effective in maintaining the engagement of autistic children [18]. In addition, it has been proved that music-based interventions have a positive effect on enhancing social skills and promoting prosocial behavior [18]. Listening to music while participating in physical activities is beneficial to physical performance as music draws the athletes’ attention to the tasks at hand [19]. After the consideration of all the factors that are important to creating an effective Taekwondo-based intervention, we worked in a multidisciplinary team and ran the intervention program in several pilot schemes. Other than the actual performance of the participants using biopsychometric tests, we would also like to examine if parents really see the difference made by the intervention. In the current study, a self-narrative approach was employed to investigate the parental perceptions of autistic children who followed Taekwondo training with music. In particular, more positive statements regarding social interactions and motor skills gains are expected.

## 2. Materials and Methods

### 2.1. Participants and Data Collection

The inclusion criteria of the study participants are parents who have at least one autistic child who participated in the intervention (Taekwondo training with elements of music therapy) for two months and have enough observation of the children’s social and physical behaviors at home or playground. The interviewees consisted of parents or caregivers (n = 13). Prior to the interview, informed consent was obtained from all participants to ensure ethical standards were upheld.

### 2.2. Characteristics of the Autistic Children in the Taekwondo Training

Chinese children aged 7–9 years old who meet the Diagnostic and Statistical Manual of Mental Disorders, Fifth Edition criteria for ASD were included in our study. However, children who (a) had received music therapy within the previous 12 months; (b) have severe sensory disorders (blindness or deafness); or (c) have underlying congenital abnormalities or other diseases that limit their abilities to engage in physical activities were excluded from our study.

Before the physical training, as assessed by the project investigator, the participants exhibited varied levels of motor skills, often facing challenges with coordination and balance. Their communication abilities differed, with some struggling more significantly in expressing needs and understanding others. Interaction with peers was minimal, and social cues were frequently neglected. Daily functioning varied widely among the 13 participants, and challenging behaviors such as repetitive actions and resistance to change were common.

### 2.3. Taekwondo Training with Music as an Intervention

The training was administered in a group setting, twice a week for 10 consecutive weeks at the Taekwondo training center close to the children’s home. Each training session lasted for 45 min and consisted of warming-up exercises, a combination of stance and hand and foot movement techniques, and cool-down exercises. The teaching syllabus followed the white belt (most junior) rank of the Taekwondo grading system with a qualified Taekwondo coach conducting the training alongside a teaching assistant and two student assistants in each session. The employed teaching assistants and student assistants were required to have a training background in sports science, physical education or rehabilitation, and it was preferable to have experience in conducting exercise training in the children population. All these individuals had attended a training workshop conducted by the investigators to ensure safety and high training quality.

The elements of music therapy were incorporated into Taekwondo training. Each piece of music was tailored according to the rhythm, dynamic, tempo, duration and associated feeling of a series of Taekwondo movements. For instance, when a forceful kicking was necessary, music with a rapid rhythm and strong, heavily accented beat would be applied. When performing breathing and cool-down exercises, the participants would follow an ascending and descending melody to perform the inspiratory and expiratory movements, respectively. The music served to guide the motor patterns in a feedforward manner. These music pieces would offer cues to the children for recurrent activities. A pedagogical training was provided to the Taekwondo coaches responsible for the teaching.

### 2.4. Self-Narrative Study through a Qualitative Interview

Semi-structured interviews (around 30 min) were utilized as the primary method of data collection in this self-narrative study. Our main aim is to investigate the perceived social skills and motor skills changes among autistic children as an outcome of Taekwondo training with elements of music therapy from the parents’ and caregivers’ perspectives.

To minimize personal bias and ensure an interpretive understanding of subjects’ own perspectives, the researchers adhered to the principles outlined [20]. This involved maintaining an open mind environment throughout the study, being receptive to the experiences and perspectives shared by the parents and giving voice to the respondents [21].

The interviews were audio-taped to allow for later access by other raters, checking for accuracy and further analysis. We transcribed the interviews, a process that involved converting the audio recordings into written text. An AI-generated text word-mining software Speechify (https://speechify.com/, accessed on 3 March 2024) was used in the conversion and text-mining. This allowed for a more thorough examination of the data among the research team members and facilitated the identification of patterns and themes in the parents’ observations and experiences by thematic analysis. The thematic analysis was conducted by identifying the existence of wordings or contents related to social skills, social intelligence, confidence, motor skills and music.

The interview questions were designed to record observations made by parents on the changes in physical ability and social communication in daily life among children with autism. Five questions introduced to build up the rapport and assess the target observations included the ability to perform simple housework and any other changes in behaviors that the parents noticed since their children had started the Taekwondo training.

Key questions:Did your children attend other types of Taekwondo training before? If yes, can you tell us more about the setting, experience and effectiveness?Have you noticed any changes in your children since he/she joined the current program?

Then, the interviewers would prompt the interviewees to report the behaviors of their children since joining the program. The study protocol was approved by the Research Ethics and Safety Approval Committee at the Lingnan University (Ref no.: EC009-2324).

## 3. Results

### 3.1. Demographic Information of Participants

A total of 13 parents or caregivers completed the interview. All the interviews were completed within 30 min as expected from our protocol. The demographic information of all interviewees is shown in Table 1. The mean age of the interviewees was 45.2 with 15% (n = 2) male and 85% (n = 11) female, respectively (Table 1). With three exceptions, most of the participants have attained the undergraduate level of education (n = 10, 76.9%).

Based on the caregivers’ statements on their observation of changes among the child participants after the intervention, the observations were grouped into two domains, namely (a) social skills and (b) motor skills. The grouping was based on the terminology and expression identified from the interview transcription.

### 3.2. Social Skills-Related Change

Social skills-related change is the most salient theme mentioned by the majority of the participants. All participating parents or caregivers (n = 13) had noticed their positive changes in social skills after the intervention, especially in the form of social reciprocity (including verbal cues and moral development) (n = 8). Some but not all the parents/caregivers noticed changes in social participation (n = 4), social intelligence (n = 4) and confidence (n = 3) after the intervention (Table 2). Children who participated in the Taekwondo training with elements of music therapy obtained different levels of social skills change.

#### 3.2.1. Social Reciprocity

The caregivers pointed out that the communal nature of the Taekwondo training sessions (group training) played a significant role in encouraging social interaction among the children. They observed that the skills acquired during these sessions were successfully applied by the children in their everyday lives. Specifically, the children were encouraged to take note of the various forms of communication occurring during the training, such as interactions between coaches, between coaches and participants, and among participants themselves. Regarding social reciprocity, particularly in terms of verbal cues, the children showed an enhanced ability to verbally express themselves and a greater understanding of the emotions of others. For example, Participant 2 said,
*“I would think of it as going to a cinema and observing a motion picture. After observing the communication between different parties; I see my child appears to have developed some tolerance for handling interaction with people. Through observation, he learned to communicate with individuals and how to deal with distinctive circumstances. The Taekwondo lesson gave him the opportunity to observe and learn.”*

Other caregivers also indicated their observation on more appropriate verbal communication among their children, who were able to express themselves via languages instead of inappropriate action. For example, Participant 12 stated,
*“In terms of communication, he (my son) expresses more of his wishes. In the past, when he did not like his younger brother taking away his belongings, he would push him away, preferring to use more actions. Now, he’ll tell his brother “don’t take my things, I’m playing with them.” Now he can express himself in a sentence instead of screaming.”*

Apart from expressing themselves efficiently, the improvement on moral development among participants was notifiable. Over 60% of caregivers (n = 8) had pointed out that their children were being more well-mannered and behaved after Taekwondo training. They believed their children benefited from the educational method of Taekwondo training. Throughout the intervention, coaches had played a significant role in giving out instructions and leading the training. The child participants were required to follow the instructions and learn to take turns during techniques practice. Caregivers pointed out that elements might be the turning point in developing moral senses among the participants. For example, Participant 11 stated,
*“After the intervention, she (my daughter) also behaved calmly while waiting in a long queue. Especially after school started, compared to the previous year, her ability to wait, queue and follow orders had improved significantly.”*

Parents also mentioned the importance of group learning on improving the discipline of their children. For example, Participant 5 stated that group learning had facilitated her children in learning to follow instructions; the result was attributed to the cooperative atmosphere created out of the teamwork partaken by the children. Moreover, children demonstrated more self-control and discipline when they were working together as a group.
*“In fact, I think group training is important to learning, because if children and adults are all working together when teaching, it will be easier for them to concentrate on learning, and his movements will be more structured.”*

In a similar manner, caregivers also indicated the Taekwondo training method as more person-centered instead of skill-centered. Engaging children in a training using a participant-centered approach has allowed the children to concentrate more on their own actions. For example, Participant 10 stated,
*“I really feel that (the children) are more obedient, he used to move around for no reason, but now he knows he can only move at the right time. For example, in a football or basketball game, you have to follow wherever the ball goes. In Taekwondo, you only need to focus on yourself. I think (this activity) is good and suitable for them.”*

#### 3.2.2. Social Skills-Related Development—Social Intelligence

Aside from social reciprocity development, enhancement in social intelligence (e.g., understanding non-verbal cues such as physical touch/empathy) among several individuals was also notifiable (n = 4). For example, Participant 2 stated that group learning played a significant role in developing social intelligence and empathy.
*“In the beginning of the lesson, he needed others to fully agree with his point of view. But now he would still express his views, but he won’t need everyone to think he is 100% right. He listened to others’ point of view as well and being more forgiving.”*

Children were found to use proper body gestures to interact with others after the intervention and demonstrate better social intelligence by taking care of others’ feelings. For example, Participant 1 indicated that her grandchild started to manage his behavior after noticing his improper body gestures may make others feel uncomfortable.
*“I am so happy that my grandchild behaves better in the public area. Before taking the class (Taekwondo training), he often touches somebody inside the metro. Those people felt offended. Now, I can see that my grandchild behaves “normal” and care about the people around.”*

Compared with physical activities in which the children previously participated, the training method of Taekwondo encourages more emergence of empathetic. Participants reported that their children were being more empathetic after observing the interaction among participants, coaches and helpers during the intervention.
*“Compared to playing basketball, he paid closer attention to others during taekwondo training and also paid close attention to what actions others were doing and how they responded to each other. Recently, when he talked to me, he was able to express himself and care about my feelings.”*

Other than that, participants exhibited prosocial behavior. A critical component of social intelligence, the skill of sharing, plays a significant role in fostering prosocial interactions. Engaging in sharing behavior further promotes the formation of friendships among those involved in the intervention. As stated, a caregiver, Participant 13, found her son becoming more sociable and developed friendships during the intervention.
*“My son starts to befriend with the other children in the class (Taekwondo training). After every class, the instructor was so nice to offer candy as a treat. My son was willing to share the candy with his classmates. I have never seen this before, and I was almost crying to see my son make friends and share things with others”.*

These excerpts illustrate that the caregivers had observed positive social skills improvement among autistic children as they were more willing and able to express themselves verbally or non-verbally in an appropriate manner and developed their social reciprocity and social intelligence skills from active social interaction during the intervention.

#### 3.2.3. Social Skills-Related Development—Confidence

Through persistent and systematic training, participants were reported to be more confident. The achievement of learning new skills and techniques, and the recognition and praise from the instructor and other students, are conducive to children’s self-confidence and self-esteem. As shared by one caregiver, Participant 11,
*“Her confidence has improved. In the past, she would feel that there were many things she could not do, and she would have a lot of worries and negative emotions before trying. But after these 20 sessions of training, she became more comfortable with each attempt and her emotion became more stable.”*

Caregivers also highlighted the significant role of the attitude of the coaches and how it affected the learning experience of the autistic children. It seems the participant is able to gain more confidence and self-esteem if the coach is sincere, non-judgmental, empathetic and respectful. Participant 2 stated,
*“My son said, “The coach is also a very nice person. Sometimes I don’t do well or I don’t understand the technique, the coach won’t stare at me or made me the spotlight (use me as a bad example) on purpose.”*
*“If the children are not able to finish the task, the coach will not give up on them but try to teach them on 1:1 basis.”*

It is found that if more patience and respect were shown by the coach, the learning experience of the children would be improved. The attitude of the person who teaches could possibly affect the learning outcome of the participants.

### 3.3. Motor Skills-Related Development

All the 13 interviewees expressed that an improvement in motor skills such as coordination and persistence of their children could be observed. The repeated movements and sequences in Taekwondo can also help autistic children develop their motor skills and coordination, as there are many balance-related techniques in the training, like kicking with one leg. As participant 7 reported,
*“My kid often fell over at home before. An observable improvement in mobility is recognized, I feel so happy that he does not get hurt from falling on the ground or colliding with tables and chairs. Even now, he can help me to take the water bottle or move some small furniture at home. Such changes are so amazing.”*

And a lot of participants also showed different degrees of advancement in persistence in physical activities. Children were able to participate for a longer time in physical activities after the intervention. As participant 3 stated,
*“His stamina has increased. For example, he will try hard to finish the task (as many kicks as possible in 2 min) but not lie down on the floor and think he can’t do it as before.”*

Children from this study enhanced motor abilities and made use of them in daily life. With a series of physical movements that require motor coordination and body control, participants tend to benefit from the intervention in terms of motor skills development.

### 3.4. Engagement of Participants

#### 3.4.1. Music-Related Elements

Several caregivers from the intervention group stated the special role music plays in engaging the participants from this study. Children were more engaged in the training when music was played; it was reported that the music provided a sense of comfort to them, and they were more focused and engaged in the training. As participant 8 illustrated,
*“Music seems to provide a sense of security to**a**utistic**children. So, I think a sense of security is helpful, especially for these children who prefer to stick with a routine.”*

In addition, music can also enhance the participants’ experience in the training. It was reported that children would be more engaged in physical activity with brisk music. Similarly, in this study, brisk music was found to engage children and make the training more fun by providing an energetic tempo and rhythms, as illustrated by the caregivers, such as Participant 12, who stated,
*“He (My son) loves music, so he will have a little more fun when music is included. For example, if the music is brisk, he will be happier and more engaged.”*

#### 3.4.2. Types of Sports and Preference

However, it was reported that a small number of participants (n = 2) did not engage in the intervention due to their own preference. The interviewees stated that their children were not particularly fond of Taekwondo training due to the type of movements involved. As participant 1 stated,
*“My girl said that she may not enjoy taekwondo very much, because girls do not like some punching and kicking movements.”*

## 4. Discussion

This self-narrative study explores if social skills and motor skills can be developed among autistic children through Taekwondo training with elements of music therapy. The results illustrated that the combination of elements in Taekwondo training with elements of music therapy promoted social skills and motor skills of autistic children by engaging them in social activities and physical activities. The current study provides preliminary findings which suggest how we can address the underdeveloped social skills and physical abilities of autistic children using Taekwondo training.

Previous studies have proved the relationship between the intensity of group activities and the amount of social interaction, which is explained by the fact that group activities provide opportunities for autistic children to practice their social skills [22]. Taekwondo training as a group activity offers chances for children to learn from direct observation. According to Social Learning Theory, children can learn through observation, imitation and modeling, and, in turn, attempt to adjust their behavior accordingly [23]. Consistent with our findings, participants have been found to have better social reciprocity skills after observing the interaction between people in the training. They were able to develop their social competence by practicing and observing. Additionally, they were more capable of expressing themselves with proper verbal cues. High predictability and high consistency of the training sessions can provide children with a sense of security and reduce their anxiety level, which is essential for enhancing their ability to engage socially [24]. Our results have shown that music (e.g., routine with brisk rhythms) can further provide a sense of security for ASD participants and increase their commitment to the training.

Apart from that, peer support was shown to encourage social participation among autistic children as they could gain mutual social support from other participants [25]. Our findings have confirmed the significant role of peer support in enhancing social reciprocity and the development of moral sense (e.g., understanding social norms). The child participants in the current study were reported to be more disciplined even in daily life events such as queuing and following orders after the intervention. According to the stages of moral development, children develop their sense of morality based on different considerations, such as punishment, self-interest, social norm, orders and contracts [26]. Before the intervention, participants from our study often demonstrated their moral development at the pre-conventional stage in which they consider their own interests first when they act. For example, it was reported that the participants tend to do whatever they want without considering other thoughts (e.g., touching others for fun on the bus or jumping the queue just because they are tired of being kept waiting). With the support from their peers, children learnt to follow rules and instructions after the intervention. Moreover, group activities created a “social norm” for the participants to follow throughout the training.

Beyond social reciprocity is the ability to interact competently and to develop a sense of morality, which is the end goal of social life [27]. It has been identified in recent studies that autistic children tend to be more socially isolated and have fewer friendships compared to their typically developing peers. This is partly because these children struggle to interpret emotions in social contexts [1]. However, through observation of interactions among various individuals, particularly between coaches and participants, it was noted that these children began to show concern for others. Remarkably, some children even exhibited prosocial behaviors (e.g., sharing) and managed to form new friendships during the intervention period.

Taekwondo training can help autistic children to build self-confidence and self-esteem [14]. Similar to our findings, autistic children were found to be sensitive to teachers’ attitudes no matter whether the teacher showed empathy to them or not [28]. It was stated that students with ASD tended to self-report as not being understood in school settings [28]. Our participants expressed that they were afraid of sports before joining the intervention as they were scolded, disrespected and teased when they took part in some physical activities previously. However, participants began to gain more confidence from our Taekwondo training and were more willing to perform their Taekwondo techniques as the coach showed respect and empathy to every one of them. It was also found that autistic children tended to have more self-efficacy and self-esteem if the teachers were able to show more empathy [28]. Similarly, our results show that a lot of caregivers noticed a great confidence boost among the participants after the intervention, suggesting the importance of the encouragement offered by the coaches.

Additionally, Taekwondo is a sport of medium to high intensity, promotes motor coordination, muscle management, and, equally important, the overall physical condition. Previous research has highlighted Taekwondo’s effectiveness as a tool for children with ASD to enhance their motor coordination skills [13,29]. Consistent with our findings, we observed marked improvements in the children’s balance and coordination abilities following their participation in the training. Furthermore, our study observed enhancements in the endurance levels of the participants.

Despite the positive impacts on autistic children, there are some limitations of this approach pending to be addressed. Taekwondo might not be a means suitable for everyone due to the combative nature of the sport; as stated in our results, some female participants did not enjoy it. Another caveat of our study is that we could only capture feedback from a limited number of participants due to the specific number of autistic children in the Taekwondo training. To address that issue, a larger sample size is required to further explore the relationships between targeted participants with different demographics and the effectiveness of the sports in improving the social skills and motor skills of a wider ASD population.

Self-selection bias might appear as the respondents were self-enrolled in this study. Self-enrollment can also cause voluntary response bias as the participants who have more information and confidence in the intervention would participate in the interview. And the data might be overrepresented due to that.

Despite these limitations, this self-narrative study has shed light on the effectiveness of Taekwondo training with elements of music therapy in alleviating social and motor skills deficits among autistic children in Hong Kong. In particular, the sport provides an inviting ground for autistic children to practice and learn how to pro-socially behave through direct observation of their peers in small group settings. Combined with the use of elements of music therapy, the intervention offers an exciting space for children to happily engage in physical activity. Moreover, the Taekwondo techniques used in the training also play an important role in enhancing the motor and physical development of autistic children. Overall, Taekwondo training with elements of music therapy shows great promise as a method for improving social and physical functioning among autistic children in Hong Kong, though a larger sample size is needed to better examine the effectiveness of the intervention.

## 5. Conclusions

Our findings show that parents observed that the Taekwondo training with elements of music therapy had a significant impact on the promotion of physical ability and social communication in autistic children.

## Figures and Tables

**Table 1 behavsci-14-00530-t001:** Demographic information.

No	Gender	Age	Relationship with Participants	Education Level
1	F	70	Grandmother	Primary School
2	F	48	Mother	Undergraduate
3	F	48	Mother	Undergraduate
4	F	43	Mother	Secondary School
5	M	43	Father	Undergraduate
6	F	43	Mother	Undergraduate
7	F	37	Mother	Undergraduate
8	F	45	Mother	Postgraduate
9	M	44	Father	Postgraduate
10	F	47	Mother	Undergraduate
11	F	40	Mother	Undergraduate
12	F	45	Mother	Undergraduate
13	F	35	Mother	Secondary School

**Table 2 behavsci-14-00530-t002:** Types of social skills changes after the intervention.

Social Skills	n (%)
Social reciprocity	8 (61.5)
Social participation	4 (30.8)
Social intelligence	4 (30.8)
Confidence	3 (23.1)
Any of the above social skills-related changes	13 (100)

## Data Availability

The data that support the findings of this study are available from the corresponding author or the first author upon reasonable request.

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
