# Peer review of "A Self-Narrative Study: Changes in Physical Ability and Social Communication in Children with Autism through Taekwondo Training with Elements of Music Therapy from the Parents’ Perspective"

_behavsci, 2024, doi:10.3390/bs14070530_

Round 1
Reviewer 1 Report
Comments and Suggestions for Authors
First, I would like to thank the authors for taking the time to conduct this research and writing this manuscript about the improvement in Physical Ability and Social Communication in 2 Children with Autism through Taekwondo training from the 3 Parent’s Perspective. This research covers a rather novel topic in autism research that could potentially benefit the ASD community. I would like to acknowledge all the time and effort authors have put into this work; however, there are many aspects of the manuscript that should be reviewed before it is considered for publication.
First, as one reads the title of the manuscript one gets the impression that this is a study that analysis the abilities and social communication of children, as well as the parental perspective on the improvements of the children in these skills. However, after reading the article it becomes obvious that this is solely about the parental perception on these improvements. This brings me to my next concern, the scientific rigor of this work. There are several aspects that should be addressed in this work: (1) the sample size, although this can be a challenge for many researchers, such a sample size makes it difficult to assess the power of the study findings, (2) the sample of children with ASD is not properly described and this is worrisome for the following reasons:
· The behavioral characteristics of the children with ASD is not reported. It is well-known that autism is a heterogenous condition and that symptoms can significantly vary from individual to individual.
· Whether participating children are clinically diagnosed with ASD and which criteria has been used to give such diagnosis.
· The intellectual, adaptive behavior, and language skills of the children with ASD is also missing, which makes it difficult to discern the main difficulties these children have.
· The gender, age, and ASD severity is also missing. These variables are also important because, for instance, research has shown gender differences amongst children with ASD.
My apprehension is that without all this information is very difficult to infer whether the results are statistically significant as to suggest there have been significant improvements in the individuals social-commutative and motor skills.
I believe that the manuscript can significantly improve if authors conduct a thorough review, addressing editorial aspects, such as grammar, punctuation, word choice, tone of voice communicating ideas, sentence/paragraph structure, etc. The organization and development of ideas is good but can be improved.
Here are some specific aspects I think should be reviewed:
Across the manuscript authors refer to ASD children, using identity first nomenclature to refer to children with ASD, however, the common, accepted terminology by the autism community is either autistic children/individuals (identity first) or children/individual with autism (person first). Using the term ASD as the identity is unusual and it also sounds odd.
In the abstract, in lines 21-23, the sentence is confusing. I suggest authors review and rephrase the sentence for more clarity.
In the introduction, lines 32-33, the definition of ASD is very narrow. Considering the complexity of the condition, a more comprehensive definition of autism should be given, taking into account how the many deficits are later addressed in the article.
Line 36, children were aged 15-21months? It could perhaps be implied, but this should be clarified.
Line 47, thought (word choice), consider changing it to a word such as intention, as it is very difficult even for neurotypicals to deduce other people´s thoughts, but intentions, on the other hand, can be inferred by physical and emotional feedback individuals provide in everyday interactions.
Line 58, what do authors mean by neuron-development? Please clarify, this expression is somewhat confusing.
Lines 61-63, avoid redundancy with the word research.
Line 76, create (word choice) considering the context of the sentence.
In line 79, author address for the first time one of the core deficits of the autism dyad, behavior. This is one of the reasons why I previously pointed out how the definition of ASD was too narrowed, as it left out this diagnostic criterion. Besides the commutation deficits, many children with autism struggle with behavioral deficits that can be detrimental to their general health and social interactions. Therefore, behavioral deficits should be addressed earlier in the text as for this statement to have meaning and significance.
Line 82, review grammar (which can make strides their social behavior).
Lines 83-85, voice tone and language use. I suggest authors review and rephrase this sentence.
Lines 93-95, objectives are not clearly stated. Review and rephrase.
Another aspect that must be reviewed is the materials used in this study. Authors refer that the primary method of data collection was a semi-structured interview; however, they fail to reference such tool. If authors created this interview for this research, this should be clearly stated, and the process of creating such instrument should also be described in detail.
In short, I suggest that authors conduct a comprehensive review of the manuscript.
Here are some references that I consider can be useful:
Hill, R. (1998). What sample size is “enough” in internet survey research? Interpersonal Computing and Technology: An Electronic Journal for the 21st Century, 6(3-4).
Isaac, S., & Michael, W. B. (1995). Handbook in research and evaluation. San Diego, CA: Educational and Industrial Testing Services.
Comments on the Quality of English LanguageI suggest authors consult the editing services of a native English speaker to review this manuscript to address grammar and fluency aspects of the paper.
Author Response
First, I would like to thank the authors for taking the time to conduct this research and writing this manuscript about the improvement in Physical Ability and Social Communication in 2 Children with Autism through Taekwondo training from the 3 Parent’s Perspective. This research covers a rather novel topic in autism research that could potentially benefit the ASD community. I would like to acknowledge all the time and effort authors have put into this work; however, there are many aspects of the manuscript that should be reviewed before it is considered for publication.
First, as one reads the title of the manuscript one gets the impression that this is a study that analysis the abilities and social communication of children, as well as the parental perspective on the improvements of the children in these skills. However, after reading the article it becomes obvious that this is solely about the parental perception on these improvements. This brings me to my next concern, the scientific rigor of this work. There are several aspects that should be addressed in this work: (1) the sample size, although this can be a challenge for many researchers, such a sample size makes it difficult to assess the power of the study findings, (2) the sample of children with ASD is not properly described and this is worrisome for the following reasons:
- The behavioral characteristics of the children with ASD is not reported. It is well-known that autism is a heterogenous condition and that symptoms can significantly vary from individual to individual.
- Whether participating children are clinically diagnosed with ASD and which criteria has been used to give such diagnosis.
- The intellectual, adaptive behavior, and language skills of the children with ASD is also missing, which makes it difficult to discern the main difficulties these children have.
- The gender, age, and ASD severity is also missing. These variables are also important because, for instance, research has shown gender differences amongst children with ASD.
Response: Thank you so much for appreciation and comments. We definitely agreed that the sample size is a concern which is common for many studies as you mentioned. The current study has already included all possible interviewees in the community project. The characteristics of the ASD children are added (Line 113-119). While the major research focus of the current self-narrative study is the observable changes on the children noticed by the parents and caregivers. It is also a big concern in the current society that the ASD support should be extended to the parents and caregivers which also be regarded as part of the service clients. The authors hope that the current study can contribute to the understanding on that research gap.
My apprehension is that without all this information is very difficult to infer whether the results are statistically significant as to suggest there have been significant improvements in the individuals social-commutative and motor skills.
I believe that the manuscript can significantly improve if authors conduct a thorough review, addressing editorial aspects, such as grammar, punctuation, word choice, tone of voice communicating ideas, sentence/paragraph structure, etc. The organization and development of ideas is good but can be improved.
Response: Thank you very much. The manuscript has been edited and revised by the research office of the university.
Here are some specific aspects I think should be reviewed:
Across the manuscript authors refer to ASD children, using identity first nomenclature to refer to children with ASD, however, the common, accepted terminology by the autism community is either autistic children/individuals (identity first) or children/individual with autism (person first). Using the term ASD as the identity is unusual and it also sounds odd.
Response: Revised as suggested. Terminology “Autistic children” is used throughout the manuscript.
In the abstract, in lines 21-23, the sentence is confusing. I suggest authors review and rephrase the sentence for more clarity.
Response: Revised as suggested. Line 22-24.
In the introduction, lines 32-33, the definition of ASD is very narrow. Considering the complexity of the condition, a more comprehensive definition of autism should be given, taking into account how the many deficits are later addressed in the article.
Line 36, children were aged 15-21months? It could perhaps be implied, but this should be clarified.
Response: Revised as suggested. Line 34-39.
Line 47, thought (word choice), consider changing it to a word such as intention, as it is very difficult even for neurotypicals to deduce other people´s thoughts, but intentions, on the other hand, can be inferred by physical and emotional feedback individuals provide in everyday interactions.
Line 58, what do authors mean by neuron-development? Please clarify, this expression is somewhat confusing.
Response: Clarified and revised as suggested. Line 59-60.
Lines 61-63, avoid redundancy with the word research.
Line 76, create (word choice) considering the context of the sentence.
Response: Revised as suggested.
In line 79, author address for the first time one of the core deficits of the autism dyad, behavior. This is one of the reasons why I previously pointed out how the definition of ASD was too narrowed, as it left out this diagnostic criterion. Besides the commutation deficits, many children with autism struggle with behavioral deficits that can be detrimental to their general health and social interactions. Therefore, behavioral deficits should be addressed earlier in the text as for this statement to have meaning and significance.
Response: Revised as suggested. Line 79-81.
Line 82, review grammar (which can make strides their social behavior).
Lines 83-85, voice tone and language use. I suggest authors review and rephrase this sentence.
Lines 93-95, objectives are not clearly stated. Review and rephrase.
Response: Revised as suggested, in the respective sentences.
Another aspect that must be reviewed is the materials used in this study. Authors refer that the primary method of data collection was a semi-structured interview; however, they fail to reference such tool. If authors created this interview for this research, this should be clearly stated, and the process of creating such instrument should also be described in detail.
Response: Thank you very much. The materials and methods section has been largely edited to clarify the intervention (tool).
In short, I suggest that authors conduct a comprehensive review of the manuscript.
Here are some references that I consider can be useful:
Hill, R. (1998). What sample size is “enough” in internet survey research? Interpersonal Computing and Technology: An Electronic Journal for the 21st Century, 6(3-4).
Isaac, S., & Michael, W. B. (1995). Handbook in research and evaluation. San Diego, CA: Educational and Industrial Testing Services.
Response: Thank you very much for the important references which is very useful for us to consider the sample size.
Comments on the Quality of English Language
I suggest authors consult the editing services of a native English speaker to review this manuscript to address grammar and fluency aspects of the paper.
Response: Thank you very much. The manuscript has been edited and revised by the research office of the university.
Reviewer 2 Report
Comments and Suggestions for Authors
I'm greatfull to review this article: Improvement in Physical Ability and Social Communication in Children with Autism through Taekwondo training from the Parent’s Perspective. This study seems very interesting but the sample is not adequate, to make a signitative statistic, only 13 partecipants in fact there is only percentage table and there is not a control group. Should be increased the sample.
There are minor grammatical mistakes.
Additional comments:
1) The main question of the research is if the Taekwondo for parental opinion is better than other sport activities to improve social skills or social and motor skill improvement is due to being together with peers.
2)There are few studies in literarture regarding the benefit of martial arts in autism spectrum disorder. Just one regarding Taekwondo in the last decade.
3)could be very useful to be combined with already known psychoeducational treatments.
4). a control group should be added and ASD sample must be increased.
5) There isn't conclusion structured paragraph.
6)References are appropriate.
7) there are not statistic consistent in table, because of a little sample. Also is missed a table with motor skill improvements after the intervention.
Author Response
I'm grateful to review this article: Improvement in Physical Ability and Social Communication in Children with Autism through Taekwondo training from the Parent’s Perspective. This study seems very interesting but the sample is not adequate, to make a singulative statistic, only 13 participants in fact there is only percentage table and there is not a control group. Should be increased the sample.
Response: Thank you so much for appreciation and comments. We definitely agreed that the sample size is a concern which is common for many studies as you mentioned. The current study has already included all possible interviewees in the community project. The characteristics of the ASD children are added (Line 113-119). While the major research focus of the current self-narrative study is the observable changes on the children noticed by the parents and caregivers. It is also a big concern in the current society that the ASD support should be extended to the parents and caregivers which also be regarded as part of the service clients. The authors hope that the current study can contribute to the understanding on that research gap.
There are minor grammatical mistakes
Response: Revised as suggested. The manuscript has been edited and revised by the research office of the university.
1) The main question of the research is if the Taekwondo for parental opinion is better than other sport activities to improve social skills or social and motor skill improvement is due to being together with peers.
Response: This question cannot be answered in the current study because we did not compare taekwondo with other sports. Importantly, taekwondo was considered as a support from previous literature.
2) There are few studies in literature regarding the benefit of martial arts in autism spectrum disorder. Just one regarding Taekwondo in the last decade.
Response: We absolutely agree that the literatures are limited. However, in the Asian culture, marital arts (such as taekwondo) are so common as a sports activity for children. Therefore, the research need is present to point out the expectations from caregivers and professional training to coaches.
3) could be very useful to be combined with already known psychoeducational treatments.
Response: Thank you for your advice.
4) a control group should be added and ASD sample must be increased.
Response: Thank you for your advice. We definitely agreed that the sample size is a concern which is common for many studies as you mentioned. The current study has already included all possible interviewees in the community project. While the major research focus of the current self-narrative study is the observable changes on the children noticed by the parents and caregivers. It is also a big concern in the current society that the ASD support should be extended to the parents and caregivers which also be regarded as part of the service clients. The authors hope that the current study can contribute to the understanding on that research gap.
5) There isn't conclusion structured paragraph.
Response: Thank you for your advice. We prepare the manuscript as suggested by the editorial office. Thank you for your understanding.
6) References are appropriate.
Response: Thank you very much.
7) there are not statistic consistent in table, because of a little sample. Also is missed a table with motor skill improvements after the intervention.
Response: Thank you for your advice. The current study would focus on the self-narrative context from the caregivers, therefore the authors decided not to put extra focus on the quantitative analysis on the motor skills improvement.
Reviewer 3 Report
Comments and Suggestions for Authors
thank you very much for inviting me to revise the following paper, The interesting topic is adequate and contemporary. Physical training is receiving more attention from researchers since it is in the treatment guidelines for ASD. Unfortunately, the current paper is problematic since most parts of the manuscript are superficial. I suggest resubmitting after relevant revisions. I have listed a series of suggestions:
- Consider a brief report than Article
2-3 (TITLE)
Parents' reports of Taekwondo training offered to children with ASD: suggestions for plan social and physical activities
34
After “movement”, please add a citation.
61-63
Since it is the main topic, you should add more citations regarding physical intervention in children and adolescents with ASD. Moreover,
in my opinion, a previous overview of martial arts is necessary, citing a paper that assessed the efficacy in ASD people
67-68
I suggest adding a website regarding the guidelines of this discipline
100
Please, add the research questions:1…,2…,3,…
METHOD
102
This sentence is appropriate after the participant section occurred
107
Furthermore, the report was the only dependent variable, while a detailed description of participants is necessary.
110
Likewise, martial training should be described, including the following sections: participants, dosage, teaching strategies, assessment, masterization of behaviors.
125-126
Too generical, rewrite, please.
132-137
I suggest a more professional table or graph explaining the content and modality of the interview as well as the scoring data collection and interpretation.
RESULT
164-165
Also, you should clarify in the method section the level and educational curriculum of the participants
168
How have you merged the different topics of the interview? it seems very subjective if you do not describe the text-mining procedure.
192
Inappropriate sentence “There was significant improvement…”
205
All the participants are better without the brackets
206
Inappropriate sentence “Result is attributed…”
208
Inappropriate sentence “Children became more disciplined…”
222-223
Inappropriate sentence, rewrite, please, you refer to a group but the report was only an individual narrative.
227
Refuse.
230
Inappropriate sentence, rewrite, please.
237
Inappropriate sentence, rewrite, please.
253-255
Inappropriate sentence, rewrite, please “significant social skills changes…”
276-277
Generally, your results are full of interpretations, please, refer to them in a described manner only.
291
Inappropriate sentence, rewrite, please, “tended to persist…”
295
Inappropriate sentence, rewrite, please, “enhanced more abilities…”
308-310
Inappropriate sentence, rewrite, please,
DISCUSSION
335-342
Inappropriate statements without empirical support.
348
Attention please, you refer to a group, but only some participants reported that experience (please, control this throughout the manuscript).
351-353
Idem like 348
357
Inappropriate sentence, rewrite, please, “after the intervention they learned…”
365-366
Inappropriate sentence, rewrite, please, “Children were more empathetic…”
386-388
Inappropriate sentence, rewrite, please: “Children demonstrate better balance…”
400-401
Inappropriate statements without empirical support: “indicate the effectiveness of…”
405-409
Idem like 401
Best luck
X.Y.
Author Response
thank you very much for inviting me to revise the following paper, The interesting topic is adequate and contemporary. Physical training is receiving more attention from researchers since it is in the treatment guidelines for ASD. Unfortunately, the current paper is problematic since most parts of the manuscript are superficial. I suggest resubmitting after relevant revisions. I have listed a series of suggestions:
Consider a brief report than Article
2-3 (TITLE)
Parents' reports of Taekwondo training offered to children with ASD: suggestions for plan social and physical activities
Response: Thank you so much for appreciation and comments. The title has been revised to be “A self-narrative study: Improvement in Physical Ability and Social Communication in Children with Autism through Taekwondo training with music therapy from the Parent’s Perspective”
34: After “movement”, please add a citation.
61-63: Since it is the main topic, you should add more citations regarding physical intervention in children and adolescents with ASD. Moreover, in my opinion, a previous overview of martial arts is necessary, citing a paper that assessed the efficacy in ASD people.
67-68: I suggest adding a website regarding the guidelines of this discipline
Response: Revised as suggested, line 113-138.
100
Please, add the research questions:1…,2…,3,…
Response: Thank you very much. After a consideration, the authors decided to keep it as a paragraph format.
METHOD
102
This sentence is appropriate after the participant section occurred
107: Furthermore, the report was the only dependent variable, while a detailed description of participants is necessary.
110: Likewise, martial training should be described, including the following sections: participants, dosage, teaching strategies, assessment, masterization of behaviors.
125-126: Too generical, rewrite, please.
Response: Revised as suggested, in the respective sentences.
132-137: I suggest a more professional table or graph explaining the content and modality of the interview as well as the scoring data collection and interpretation.
Response: Thank you very much. After a consideration, the authors decided to keep it as a paragraph format.
RESULT
164-165: Also, you should clarify in the method section the level and educational curriculum of the
Response: Revised as suggested, line 113-138.
168: How have you merged the different topics of the interview? it seems very subjective if you do not describe the text-mining procedure.
192: Inappropriate sentence “There was significant improvement…”
205: All the participants are better without the brackets
206: Inappropriate sentence “Result is attributed…”
208: Inappropriate sentence “Children became more disciplined…”
222-223: Inappropriate sentence, rewrite, please, you refer to a group but the report was only an individual narrative.
227: Refuse.
230:Inappropriate sentence, rewrite, please.
237: Inappropriate sentence, rewrite, please.
253-255: Inappropriate sentence, rewrite, please “significant social skills changes…”
276-277: Generally, your results are full of interpretations, please, refer to them in a described manner only.
291: Inappropriate sentence, rewrite, please, “tended to persist…”
295: Inappropriate sentence, rewrite, please, “enhanced more abilities…”
308-310: Inappropriate sentence, rewrite, please,
Response: Revised as suggested, in the respective sections and sentences.
DISCUSSION
335-342:Inappropriate statements without empirical support.
348: Attention please, you refer to a group, but only some participants reported that experience (please, control this throughout the manuscript).
351-353: Idem like 348
Response: Thank you very much for your comment. We have revised the discussion and inserted more reference to enhance the empirical support. Besides, the current self-narrative study is important to state and explain the observation for the interviewees, i.e. the parents and caregivers of the children with autism.
357: Inappropriate sentence, rewrite, please, “after the intervention they learned…”
365-366: Inappropriate sentence, rewrite, please, “Children were more empathetic…”
386-388: Inappropriate sentence, rewrite, please: “Children demonstrate better balance…”
400-401: Inappropriate statements without empirical support: “indicate the effectiveness of…”
405-409: Idem like 401
Response: Revised as suggested, in the respective sentences.
Round 2
Reviewer 1 Report
Comments and Suggestions for Authors
The authors addressed all concerns and suggestions pointed out on the first review, which I believe improved the quality of the manuscript. I would suggest they do a final, detailed review focusing on English grammar, as it is the only aspect that could still be improved as in some sections of the article the grammar use interferes with the fluency of the paper. Besides that, I would not make any other remarks.
Comments on the Quality of English LanguageThe English quality of the manuscript has significantly improved. I suggest the authors review the article one last time as there are instances where the grammar interferes with the clarity of the sentences (mostly verb tenses issues). All in all, I would like to acknowledge the time and effort authors took to address all the suggestions made in the review.
Author Response
Thank you very much. The manuscript has been proofread by the university's English editing service again.
Reviewer 2 Report
Comments and Suggestions for Authors
The sample is always small, but the motivation was explained by the authors, explaining that the article is narrative in nature.
The text of the manuscript has improved and the authors have answered all the requested questions.
I believe that the manuscript is interesting from parents point of view and that it develops new useful possibilities for these children, to be added to psychotherapeutic treatments
Author Response
Response: Thank you very much. Your support is a strong motivation for us.
Reviewer 3 Report
Comments and Suggestions for Authors
I have agreed to revise your paper, having a more complete vision of your research. I have suggested another major revision before the consideration. Let me list the following changes.
Introduction
In the introduction section, the order of citations seems not to conform.
42 autistic children vs. Autistic children
44 [1–7]? I have counted minor citations before.
51 autistic children vs. Autistic children (please revise throughout the manuscript)
64-68
According to a recent study, participating in physical activities can reduce motor abnormalities in Autistic children [11]. Another study highlighted the importance of regular physical activities in addressing abnormalities [12]. Physical activities, especially martial arts-related interventions were found to be effective in improving the social skills and motor skills (postural control) of Autistic children [14]
Since this should be the main literature previously your proposal training, you need to describe more deeply the method and result of the related studies. This is fundamental since you should connect your investigation with similar studies.
76-80
Considering the combination of different motor techniques, practicing Taekwondo could improve balancing and physical ability in Autistic children [14]. The practice of Taekwondo kicks and punches requires a high level of physical coordination and control as many actions demand balancing on one side of the body. The rehashed developments and arrangements in Taekwondo can offer assistance to Autistic children to create their body control and coordination abilities.
Once again, what study regarding martial arts has demonstrated an increment in motor and social abilities in the clinical sample? Intellectual disabilities? Autism? Meanwhile, the topic is interesting but lacks still of focused literature review.
98-105
The study aims to investigate the social and motor benefits of Taekwondo training in children with ASD via parental reports.
Vs.
The study aims to report the perceived benefits of our program among the parents of the participating children.
As the key stakeholders, parental support plays an important role in the sustainability of the intervention (e.g. parents agree to sign up, bring the children to the venue and bring them home, follow up with the assigned exercise between sessions, etc). Also, we would like to systematically obtain feedback from parents and caregivers who will provide valuable insights into the improvement of sport-based intervention. A self-narrative approach is employed to investigate the observations from the parents and caregivers.
This part is redundant here, you should indicate the following research questions.
109
(b) have enough observation of the children’s social and physical behaviors at home or playground, and
the caregiver knows per se of the child, redundant.
154
This allowed for a more thorough amination of the data among the research team members and facilitated the identification of patterns and themes in the parents’ observations and experiences via thematic analysis.
Thank you for having better described the training section, but to understand your data extraction, you should explain the thematic analysis that you applied.
165
autism vs Autism
187-188
autistic vs Autistic children.
202
Social reciprocity: Social competence (verbal cues) 8 (61.5)
Social participation: (group learning) 4 (30.8)
…
In the method section, you need to explain how you have extracted these domains for the reader to understand the results. For example, you could gather data via a ratio of positive and negative terms stated, a text mining of co-occurrences or frequency, a multidimensional analysis, and so on.
Generally, since you have only 12 reports, consider a table of synthesis of interviews comparing the same table of experimental and control group
205
Refuse
RESULTS
Your result section includes social skills, motor, and music experience. Nevertheless, it should be connected with the interview, while you have described subjectively only the last question.
1. Can you please tell me your understanding of Taekwondo? 168
2. Can you please list out the characteristics of individuals with Autism? 169
3. What do you think about the feasibility of Taekwondo in children with autism? 170
4. Did your children attend other types of Taekwondo training before? If yes, can 171
you tell us more about the setting, experience, and effectiveness? 172
5. Have you noticed any changes in your children since he/she joined the current 173
program?
In my opinion, you should gather better your data showing them in a more correct manner.
Discussion
This section is full of surrounding theoretical statements but it is not connected with your concrete results. Likewise, without a better organization of your collection and analysis of data you could not interpret them.
To sum up, I think that your proposal although interesting and original needs to be revised since my previous scientific doubts have not been still sufficiently addressed. This paper should emphasize the training dedicated and connected literature, showing first interesting reports and future developments.
Mainly issues: literature review, aim and research question, data collecting and outcomes, discussion, and citations.
I hope you can dedicate more time to following my suggestions.
The professional exchange is the key to the improvement of the manuscript unless it will remain hardly replicable and remarked.
Good Luck
X.Y.
Author Response
Reviewer 3
I have agreed to revise your paper, having a more complete vision of your research. I have suggested another major revision before the consideration. Let me list the following changes.
Introduction
In the introduction section, the order of citations seems not to conform.
42 autistic children vs. Autistic children
4 [1–7]? I have counted minor citations before.
51 autistic children vs. Autistic children (please revise throughout the manuscript)
Response: Sorry for the mistake. The changes (autistic children) have been made throughout the manuscript.
64-68
According to a recent study, participating in physical activities can reduce motor abnormalities in Autistic children [11]. Another study highlighted the importance of regular physical activities in addressing abnormalities [12]. Physical activities, especially martial arts-related interventions were found to be effective in improving the social skills and motor skills (postural control) of Autistic children [14]
Since this should be the main literature previously your proposal training, you need to describe more deeply the method and result of the related studies. This is fundamental since you should connect your investigation with similar studies.
Response: Thank you very much for your advice. The paragraph has been revised as follows.
Line 58-69: Aside from social skills deficits, autistic children often face challenges with motor coordination and balancing. For example, researchers found that these children had more difficulties in maintaining stability in terms of postural and coordination within the media-lateral [9]. Limitations in gross and fine motor skills may lead to delayed motor development in autistic children [10]. Participation in physical activities is a direct way to develop and enhance gross and fine motor skills. In a recent study, performing regular exercises in physical education classes can reduce motor abnormalities in autistic children, tapped with the Test of Gross Motor Development Second Edition (TGMD-2) [11]. Another study highlighted the importance of regular physical activities in addressing motor abnormalities in children with autism-like characteristics. This was achieved through a school-based fundamental-motor-skill intervention that included underhand rolling, overhand throwing and other basic movement training [12].
76-80
Considering the combination of different motor techniques, practicing Taekwondo could improve balancing and physical ability in Autistic children [14]. The practice of Taekwondo kicks and punches requires a high level of physical coordination and control as many actions demand balancing on one side of the body. The rehashed developments and arrangements in Taekwondo can offer assistance to Autistic children to create their body control and coordination abilities.
Once again, what study regarding martial arts has demonstrated an increment in motor and social abilities in the clinical sample? Intellectual disabilities? Autism? Meanwhile, the topic is interesting but lacks still of focused literature review.
Response: Thank you very much for your advice. The paragraph has been revised as follows.
Line 70-89: To cater for the needs of autistic children, a sport that enhances social interaction opportunities and techniques for improving motor skills would be ideal. Taekwondo, a traditional Korean martial art, requires various movements, including strikes, kicks, and blocks. Taekwondo kicks and punches require a high level of physical coordination and control as many moves demand balancing on one side of the body, the constant practice of these moves is likely to promote daily life gross motor functioning can help autistic children to create their body control and coordination abilities. Previous research has suggested that Taekwondo training enhances balance control in autistic children compared to those who did not receive any intervention [14]. Moreover, one of the key aspects of Taekwondo is the emphasis on following instructions, discipline, and maintaining self-control, which may enhance children’s social abilities [13]. The rules-bounded nature of Taekwondo training gives autistic children the opportunity to understand and practice social rules, which can make strides in their social behaviors [16]. The discipline and self-control required in Taekwondo can also help children with autism manage their behaviors more effectively. This will lead to advancements in their social intelligence and making connections with peers. Meanwhile, improving motor skills has provided a solid ground for autistic children to improve their socio-communication skills promoting and generalizing the skills learned in therapeutic settings [15]. The review also summarized that the effectiveness of individual sports in improving motor and social skills was demonstrated in the ASD population [15].
98-105
The study aims to investigate the social and motor benefits of Taekwondo training in children with ASD via parental reports.
Vs.
The study aims to report the perceived benefits of our program among the parents of the participating children.
As the key stakeholders, parental support plays an important role in the sustainability of the intervention (e.g. parents agree to sign up, bring the children to the venue and bring them home, follow up with the assigned exercise between sessions, etc). Also, we would like to systematically obtain feedback from parents and caregivers who will provide valuable insights into the improvement of sport-based intervention. A self-narrative approach is employed to investigate the observations from the parents and caregivers.
This part is redundant here, you should indicate the following research questions.
Response: Thank you very much for your advice. The paragraph has been revised as follows.
Line 99-104: While it is important to measure the actual performance of the participants using biopsychometric tests, we would also like to examine if parents really see the difference made by the intervention. A self-narrative approach was employed to investigate the parent-observed performance of autistic children. In the current study, social and motor skills development induced specifically by our Taekwondo training is the key investigative focus.
109
(b) have enough observation of the children’s social and physical behaviors at home or playground, and the caregiver knows per se of the child, redundant.
Response: Thank you very much for your advice. The paragraph has been revised as follows.
Line 107-110: The inclusion criteria of the study participants are parents who have at least one autistic child who participated in the intervention (Taekwondo training with elements of music therapy) for two months and have enough observation of the children’s social and physical behaviours at home or playground.
154
This allowed for a more thorough amination of the data among the research team members and facilitated the identification of patterns and themes in the parents’ observations and experiences via thematic analysis.
Thank you for having better described the training section, but to understand your data extraction, you should explain the thematic analysis that you applied.
Response: Thank you very much for your advice. The paragraph has been revised as follows.
Line 147-151: This allowed for a more thorough examination of the data among the research team members and facilitated the identification of patterns and themes in the parents’ observations and experiences via thematic analysis. The thematic analysis was done by identifying the existence of wordings or contents related to social skills, social intelligence, confidence, motor skills and music.
165
autism vs Autism
187-188
autistic vs Autistic children.
Response: Sorry for the mistake. The changes (autistic children) have been made throughout the manuscript.
202
Social reciprocity: Social competence (verbal cues) 8 (61.5)
Social participation: (group learning) 4 (30.8)
…
In the method section, you need to explain how you have extracted these domains for the reader to understand the results. For example, you could gather data via a ratio of positive and negative terms stated, a text mining of co-occurrences or frequency, a multidimensional analysis, and so on.
Generally, since you have only 12 reports, consider a table of synthesis of interviews comparing the same table of experimental and control group
Response: Thank you very much for your advice. After a discussion among the authors, we agreed to rewrite the paragraph and table 2. The types of social skill improvements are now matching with the thematic analysis. The methodological details are supplemented in the method part. Thank you very much.
RESULTS
Your result section includes social skills, motor, and music experience. Nevertheless, it should be connected with the interview, while you have described subjectively only the last question.
- Can you please tell me your understanding of Taekwondo? 168
- Can you please list out the characteristics of individuals with Autism? 169
- What do you think about the feasibility of Taekwondo in children with autism? 170
- Did your children attend other types of Taekwondo training before? If yes, can 171
you tell us more about the setting, experience, and effectiveness? 172
- Have you noticed any changes in your children since he/she joined the current 173
program?
In my opinion, you should gather better your data showing them in a more correct manner.
Response: Thank you very much for your advice. The current presentation is confusing, therefore we decided to rewrite the part in order to avoid any sidetrack on the methodology.
Line: 152-159: The interview questions were designed to record observations made by parents on the changes in physical ability and social communication in daily life among children with autism. The target observations included the ability to perform simple housework and any other changes in behaviors that the parents noticed since their children had started the Taekwondo training. During the interview, the participants were asked to mention their understanding of Taekwondo and Autism as a familiarization. Then, the major focus would go to the notification changes in their children since joining the program.
Discussion
This section is full of surrounding theoretical statements but it is not connected with your concrete results. Likewise, without a better organization of your collection and analysis of data you could not interpret them.
To sum up, I think that your proposal although interesting and original needs to be revised since my previous scientific doubts have not been still sufficiently addressed. This paper should emphasize the training dedicated and connected literature, showing first interesting reports and future developments.
Mainly issues: literature review, aim and research question, data collecting and outcomes, discussion, and citations.
I hope you can dedicate more time to following my suggestions.
The professional exchange is the key to the improvement of the manuscript unless it will remain hardly replicable and remarked.
Response: Thank you very much for your comment and advice. We appreciate your comments and suggestions. Your professional comments are very useful and make them think about the future direction of the research project. The current self-narrative study stated the observations from parents and caregivers which were not investigated comprehensively before. Thank you very much for giving us a chance to improve and excel.
Round 3
Reviewer 3 Report
Comments and Suggestions for Authors
Thank you for replying point-to-point.
I list various suggestions:
78-89
....to those who did not receive any intervention [14]. Moreover, one of the key aspects of Taekwondo is the emphasis on following instructions, discipline, and maintaining self-control, which may enhance children’s social abilities [13].
[16]. The discipline and self-control required in Taekwondo can also help children with autism manage their behaviors more effectively. This will lead to advancements in their social intelligence and improve the skills learned in therapeutic settings [15].
Please check the order of citations.
99-104
While it is important to measure the actual performance of the participants using bio-psychometric tests, we would also like to examine if parents see the difference made by the intervention. A self-narrative approach was employed to investigate the parent-observed performance of autistic children. In the current study, social and motor skills development induced specifically by our Taekwondo training is the key investigative focus.
I have a problem with this part. The research questions are still not corrected.
I add some examples:
The study aimed to investigate parental perceptions of 13 ASD children who followed Taekwondo training with music. In particular, more positive statements regarding social interactions and balance gains (qualitatively measurable) are expected.
METHOD
114-120
Characteristics of the autistic children in the Taekwondo training Chinese children aged 7-9 years old who meet the Diagnostic and Statistical Manual of Mental Disorders, Fifth Edition criteria for ASD were included in our study. However, children who (a) Have received music therapy within the previous 12 months; (b) have severe sensory disorders (blindness or deafness), or (c) have underlying congenital abnormalities or other diseases that limit their abilities to engage in physical activities were excluded from our study.
It would help if you extended this section, including a general description of the skills related to participants before the training (motor, communication, interaction, functioning, challenging behaviors).
122
2.3 Taekwondo training as an intervention
You also need to include the music as an aspect of your training
137
Semi-structured interviews (around 30 minutes) were utilized as the primary method.
You need to include the number of questions and domains, along with examples of interviews and responses.
150-151
The thematic analysis was done by identifying the existence of wordings or contents related to social skills, social intelligence, confidence, motor skills, and music.
As suggested in the previous revision, you should apply a manualized content/text analysis using text mining software. Currently, the methodology and results are still uncorrected.
Moreover, social skills and intelligence, confidence, motor skills, and music domains refer to a subjective assessment of risk content validity.
171-174
Based on the caregivers’ statements on their observation of changes among the child participants after the intervention, two main themes were identified regarding the major areas of improvement, namely a) social skills and b) motor skills.
“Two main themes were identified”
How have you defined and collected your domains?
These improvements can provide evidence of how taekwondo training with elements of music therapy made a positive impact on autistic children.
“improvements” need a pre-post assessment; you could show the results in a positive perception of training based on behaviors displayed.
178-184
Social skills-related improvement is the most salient theme mentioned by the majority of the participants. All participating parents or caregivers (n=13) had noticed a positive change in social skills after the intervention, especially in the form of social reciprocity (including verbal cues and moral development) (n=8). Some but not all the parents/caregivers noticed a change in social participation (n=4), social intelligence (n=4), and confidence (n=3) after the intervention (Table 2). Children who participated in the taekwondo training with elements of music therapy obtained different levels of social skills improvement.
Idem, the previous doubts and suggestions are not being addressed.
Subparagraphs of results.
The selection of text regarding results seems subjective, you need before applying a methodology of data collection but also of definition of your domains. Generally, since you gathered an interview, I suggest limiting the domains of observations cause problems of content validity above mentioned.
Introduction
Please, consider the following studies to extend your related literature
Lang, R., Koegel, L. K., Ashbaugh, K., Regester, A., Ence, W., & Smith, W. (2010). Physical exercise and individuals with autism spectrum disorders: A systematic review. Research in Autism Spectrum Disorders, 4(4), 565-576.
Sorensen, C., & Zarrett, N. (2014). Benefits of physical activity for adolescents with autism spectrum disorders: A comprehensive review. Review Journal of Autism and Developmental Disorders, 1, 344-353.
Sowa, M., & Meulenbroek, R. (2012). Effects of physical exercise on autism spectrum disorders: A meta-analysis. Research in autism spectrum disorders, 6(1), 46-57.
Wang, S., Chen, D., Yoon, I., Klich, S., & Chen, A. (2022). Bibliometric analysis of research trends of physical activity intervention for autism spectrum disorders. Frontiers in Human Neuroscience, 16, 926346.
Healy, S., Nacario, A., Braithwaite, R. E., & Hopper, C. (2018). The effect of physical activity interventions on youth with autism spectrum disorder: A meta‐analysis. Autism Research, 11(6), 818-833.
Reinders, N. J., Branco, A., Wright, K., Fletcher, P. C., & Bryden, P. J. (2019). Scoping review: Physical activity and social functioning in young people with autism spectrum disorder. Frontiers in Psychology, 10, 120.
Huang, J., Du, C., Liu, J., & Tan, G. (2020). Meta-analysis on intervention effects of physical activities on children and adolescents with autism. International journal of environmental research and public health, 17(6), 1950.
Toscano, C. V., Ferreira, J. P., Quinaud, R. T., Silva, K., Carvalho, H. M., & Gaspar, J. M. (2022). Exercise improves the social and behavioral skills of children and adolescents with autism spectrum disorders. Frontiers in Psychiatry, 13, 1027799.
Bell, A., Palace, K., Allen, M., & Nelson, R. (2016). Using martial arts to address social and behavioral functioning in children and adolescents with autism spectrum disorder. Therapeutic Recreation Journal, 50(2), 176-180.
Phung, J. N., & Goldberg, W. A. (2021). Mixed martial arts training improves social skills and lessens problem behaviors in boys with Autism Spectrum Disorder. Research in Autism Spectrum Disorders, 83, 101758.
Li, L., Li, H., Zhao, Z., & Xu, S. (2022). Comprehensive intervention and effect of martial arts routines on children with autism. Journal of Environmental and Public Health, 2022.
Greco, G., & De Ronzi, R. (2020). Effect of karate training on social, emotional, and executive functioning in children with autism spectrum disorder. Journal of Physical Education and Sport, 20(4), 1637-1645.
Hosokawa, K., Yano, N., & Sumimoto, A. (2024). Scoping Review of Martial Arts Intervention Studies for Autism Spectrum Disorders Scoping Review of Martial Arts for ASD. International Journal of Sport and Health Science, 202320.
Rivera, P., Renziehausen, J., & Garcia, J. M. (2020). Effects of an 8-week Judo program on behaviors in children with Autism Spectrum Disorder: A mixed-methods approach. Child Psychiatry & Human Development, 51, 734-741.
Bo, J., Pang, Y., Dong, L., Xing, Y., Xiang, Y., Zhang, M., ... & Shen, B. (2019). Brief report: Does social functioning moderate the motor outcomes of a physical activity program for children with autism spectrum disorders—A pilot study. Journal of autism and developmental disorders, 49, 415-421.
To conclude, I think that the martial arts topic is in line with exercise for the ASD population and that your topic is original. Nevertheless, your research process needs to be revised.
The following items need to be changed:
- Introduction (meliorating and extending)
- Research questions (specific and measurable)
- Method: participants (currently incomplete)
- Define the domains you studied
- Defining better the training (have data from direct training?)
- Interview and data extraction
- Results (reformulate)
- Discussion (reformulate)
Once again, I hope my clarification helps you in your research.
Good Luck
Author Response
We greatly appreciate your feedback and recommendations. Your insights have undoubtedly enhanced the quality and clarity of our manuscript, making it a more accurate representation of our work. In this revision, we have specifically addressed the research question and methodology for greater clarity. Following your advice on references, we have incorporated the most recent studies from your suggestions. We are truly thankful for your assistance. It is our hope that this revised version meets the expected standards. Thank you once again.